# Chitosan-Stabilized Lipid Vesicles with Indomethacin for Modified Release with Prolonged Analgesic Effect: Biocompatibility, Pharmacokinetics and Organ Protection Efficacy

**DOI:** 10.3390/pharmaceutics17040523

**Published:** 2025-04-16

**Authors:** Angy Abu Koush, Eliza Gratiela Popa, Beatrice Rozalina Buca, Cosmin Gabriel Tartau, Iulian Stoleriu, Ana-Maria Raluca Pauna, Liliana Lacramioara Pavel, Paula Alina Fotache, Liliana Mititelu Tartau

**Affiliations:** 1Department of Pharmacology, Faculty of Medicine, ‘Grigore T. Popa’ University of Medicine and Pharmacy, 700115 Iasi, Romania; maierean_angy@yahoo.com (A.A.K.); beatrice-rozalina.buca@umfiasi.ro (B.R.B.); lylytartau@yahoo.com (L.M.T.); 2Department of Pharmaceutical Technology, Faculty of Pharmacy, ‘Grigore T. Popa’ University of Medicine and Pharmacy, 700115 Iasi, Romania; 3Department of Histology, Faculty of Medicine, ‘Grigore T. Popa’ University of Medicine and Pharmacy, 700115 Iasi, Romania; cosmin.tartau@gmail.com; 4Faculty of Mathematics, ‘Alexandru Ioan Cuza’ University, 700506 Iasi, Romania; iulian.stoleriu@uaic.ro; 5Department of Anatomy, Faculty of Medicine, ‘Grigore T. Popa’ University of Medicine and Pharmacy, 700115 Iasi, Romania; paunaanamariaraluca@gmail.com; 6Medical Department, Faculty of Medicine and Pharmacy, ‘Dunarea de Jos’ University, 800010 Galati, Romania; doctorpavel2012@yahoo.com; 7Department of Morphological and Functional Sciences, Faculty of Medicine and Pharmacy, ‘Dunarea de Jos’ University, 800010 Galati, Romania; fotache.paula@yahoo.com

**Keywords:** indomethacin, lipid vesicles, biocompatibility, tail flick, mice

## Abstract

**Background/Objectives**: Indomethacin (IND) is a widely used non-steroidal anti-inflammatory drug (NSAID) effective in managing pain and inflammation. However, its therapeutic use is often limited by gastrointestinal irritation and low bioavailability. This study aimed to evaluate the biocompatibility, release kinetics, and analgesic potential of IND-loaded chitosan (CHIT)-stabilized lipid vesicles (IND-ves) in comparison to free IND, focusing on their in vivo effects and impact on somatic nociceptive reactivity in mice. **Methods**: IND-ves were prepared using a molecular droplet self-assembly technique, followed by CHIT coating to enhance stability and control drug release. Mice were administered either free IND or IND-ves, and various physiological parameters, including liver and kidney function, oxidative stress markers, immune cell activity, and histopathological changes in key organs, were assessed. Plasma drug release kinetics and analgesic effects were evaluated using the tail-flick test. **Results**: Both IND and IND-ves demonstrated good biocompatibility, with no significant changes in hematological, biochemical, or immunological profiles. IND-ves exhibited a sustained release profile, with drug release initiating at 30 min and peaking at 3 h, while free IND displayed a rapid release and potential gastric mucosal damage. IND-ves did not induce oxidative stress or inflammation and maintained organ integrity, particularly protecting against gastric injury. Additionally, the prolonged release profile of IND-ves contributed to extended analgesic effects in the tail-flick test. **Conclusions**: CHIT-stabilized lipid vesicles offer a promising drug delivery system for IND, enhancing drug release, prolonging analgesic efficacy, and minimizing gastrointestinal irritation. These findings suggest that IND-ves could serve as a safer and more effective alternative for NSAID therapy.

## 1. Introduction

Modern drug delivery strategies aim to design carrier systems that align with a detailed understanding of interactions within the biological environment, considering factors such as the target cell population, receptor dynamics, the mechanism of action of active substances, pharmacokinetics, site of action, molecular mechanisms, and the underlying pathobiology of diseases [1,2]. These efforts have gained prominence in pharmaceutical research over recent decades as means to overcome limitations in drug bioavailability, therapeutic efficacy, and patient adherence. Among these strategies, polymeric drug delivery systems have emerged as promising tools [3,4]. Composed of biodegradable and biocompatible polymers, these systems enable controlled drug release, improve solubility, enhance absorption rates, and stabilize formulations [5,6,7]. By reducing the required dosage for therapeutic effectiveness, they mitigate adverse effects and foster prolonged therapeutic action, distinguishing them from conventional formulations [8,9].

The incorporation of nanotechnology in drug delivery has further revolutionized the field. Nanoparticles, such as lipid-based systems and liposomes, capitalize on the unique physicochemical properties of nanomaterials, including their small size, altered pharmacokinetics, and ability to cross biological barriers [2,10,11,12]. These systems are particularly effective in addressing challenges like poor solubility, low bioavailability, and high toxicity associated with traditional drugs [13,14]. Liposomes are versatile lipid-based nanoparticles which have gained wide clinical acceptance for their ability to deliver both hydrophobic and hydrophilic molecules, including small drugs, proteins, and nucleic acids [15,16]. They enhance pharmacodynamic effects, reduce adverse reactions, and improve the detection and treatment of diseases [17].

One of the most prevalent applications of these advanced delivery systems lies in enhancing the safety and efficacy of non-steroidal anti-inflammatory drugs (NSAIDs) [18,19]. NSAIDs, widely used for their analgesic, antipyretic, and anti-inflammatory properties, achieve their effects primarily by inhibiting cyclooxygenase (COX) enzymes, which convert arachidonic acid into prostaglandins that mediate inflammation, pain, and fever [20]. However, the non-selective inhibition of both COX-1 and COX-2 isoforms is associated with adverse effects, such as gastrointestinal toxicity, peptic ulcers, and renal complications, especially during prolonged use [21,22]. Nanoparticle-based delivery systems provide an innovative approach to minimize these side effects by enabling modified release, reducing peak plasma concentrations, and shielding gastrointestinal tissues from direct exposure to the drug [19,23]. This approach significantly improves therapeutic outcomes, particularly for conditions characterized by chronic inflammation and pain [18,24].

Indomethacin (IND), a potent NSAID, illustrates the advantages of nanoparticle-based delivery systems. Chemically designated as 1-(4-chlorobenzoyl)-5-methoxy-2-methyl-1H-indole-3-acetic acid, it is identified with the molecular formula of C19H16ClNO4 and a molecular weight of 357.79 g/Mol. It features a unique structure with a methoxy group, a chlorobenzoyl moiety, and an indole core [25]. As a weak acid (pKa ~ 4.5) and moderately lipophilic compound (Log P ~ 4.3), IND is poorly soluble in water but exhibits solubility in organic solvents such as ethanol, DMSO, and methanol [26,27]. Its amphiphilic nature, derived from its carboxylic acid group, facilitates interactions in both aqueous and lipid environments, making it an ideal candidate for nanoparticle-based formulations [28].

Pharmacologically, IND is rapidly absorbed from the gastrointestinal tract, with nearly 100% bioavailability following oral administration [29]. Its pharmacokinetics is characterized by high plasma protein binding (~99%, primarily to albumin) and extensive distribution, crossing the blood–brain barrier and placenta [30,31]. It undergoes a biphasic elimination process, with a shorter initial half-life (~1 h) and a longer secondary phase ranging from 2.6 to 11.2 h [32]. Hepatic metabolism plays a crucial role, involving processes such as glucuronidation, O-demethylation, and N-deacylation, predominantly mediated by cytochrome P450 enzymes (CYP2C9). The resulting metabolites are excreted via urine and feces [29,33,34].

The mechanism of action of IND involves potent inhibition of COX-1 and COX-2 enzymes, leading to a reduction in prostaglandin synthesis. This suppresses the inflammatory response, alleviates pain, and reduces fever [29,35]. Clinically, it is utilized for various inflammatory conditions, including rheumatoid arthritis, osteoarthritis, ankylosing spondylitis, and gout [36,37]. It is also employed to close patent ductus arteriosus in preterm infants [38]. However, its strong inhibition of COX-1 predisposes patients to gastrointestinal side effects such as dyspepsia, ulcers, and bleeding [34,39]. Renal and cardiovascular risks further complicate its long-term use, as decreased prostaglandin production can impair renal function and increase thrombotic events [40,41,42,43].

Incorporating IND into nanoparticle systems addresses these limitations by achieving modified release and reducing direct exposure to the gastrointestinal mucosa. Nanoparticles also prolong therapeutic effects, enhance anti-inflammatory efficacy, and improve intracellular drug concentrations in experimental models [44]. For instance, lipid-based nanocarriers and polymeric nanoparticles have shown promise in reducing gastrointestinal irritation and providing sustained drug delivery [45,46]. However, despite promising in vitro results, challenges remain in achieving consistent therapeutic effects in vivo, particularly in the treatment of inflammatory diseases [47].

Future research needs to focus on optimizing nanoparticle formulations to enhance bioavailability, ensure stability, and achieve targeted delivery. Overcoming these challenges will pave the way for safer and more effective NSAID therapies, with IND serving as a model compound for integrating advanced drug delivery technologies into clinical practice. By addressing the pharmacokinetic and pharmacodynamic complexities of such drugs, these innovative systems hold the potential to revolutionize the management of inflammation and pain across diverse patient populations.

The objective of this study was to evaluate the biocompatibility, release kinetics, and analgesic potential of indomethacin-loaded chitosan-stabilized lipid vesicles (IND-ves) in comparison to free IND. Specifically, we sought to determine the pharmacokinetic advantages of IND-ves, assess its impact on somatic nociceptive reactivity in mice, and investigate its effects on physiological parameters such as liver and kidney function, oxidative stress markers, and immune cell activity. Through this comprehensive evaluation, our aim was to establish whether the IND-ves formulation could offer a more effective and safer alternative for NSAID therapy, particularly by enhancing drug release, prolonging analgesic efficacy, and minimizing gastrointestinal irritation.

## 2. Materials and Methods

### 2.1. Substances

The substances used for nanoparticle preparation were provided by Sigma-Aldrich Chemical Co. (Steinheim, Germany): chitosan (CHIT—catalog code C3646, derived from crab shells, molecular weight of 310,000 g/Mol, 80% degree of N-deacetylation; polydispersity index of 3.26); IND (catalog code I7378, 98.5% purity, molecular weight of 357.79 g/Mol); phosphatidylcholine (catalog code P5638, type II-S, soy-derived, containing 14–29% choline); cholesterol (catalog code C8667, ≥99% purity, molecular weight of 386.65 g/Mol); chloroform (catalog code C2432, stabilized with 100–200 ppm amylene, ≥99.5% purity, molecular weight of 119.38 g/Mol); glacial acetic acid (catalog code 695092, ≥99.7% purity, molecular weight of 60.05 g/Mol); ethanol (catalog code E7148, 95% purity, molecular weight of 46.07 g/Mol).

### 2.2. Preparation of Lipid Vesicles Encapsulating IND

The IND-loaded lipid vesicles were prepared using a molecular droplet self-assembly method [32]. Ethanol served as the solvent for lipids, cholesterol, and IND, while CHIT was dissolved in 0.5% glacial acetic acid. To ensure uniform dispersion, 0.09 g soy-derived phosphatidylcholine was first dissolved in 0.66 mL ethanol. Cholesterol (0.015 g) and IND (0.01 g) were dissolved separately in 1 mL of ethanol each. The IND solution was then combined with the lipid solution, bringing the final volume to 1.66 mL. The mixture was sonicated using a Bandelin SONOPULS ultrasonic homogenizer (Berlin, Germany) under controlled amplitude and temperature. Ultrasound treatment was performed at 25% amplitude for 10 min at 29 °C, generating 20,000 kJ of energy, with pulse cycles of 5 s on and 2 s off to ensure uniform dispersion. The process was conducted in an ice-cooled water bath to prevent overheating and degradation of the components. This technique facilitated the conversion of multilamellar vesicles into unilamellar ones with a well-defined size and morphology. The resulting sonicated mixture was added to 8.33 mL double-distilled water at room temperature, triggering the spontaneous self-assembly of IND-loaded lipid vesicles and yielding a translucent dispersion. For CHIT-coated vesicles, the ethanol-based lipid–IND solution (1.66 mL) was added dropwise to 8.33 mL of 0.25% CHIT solution under magnetic stirring. The stirring process was maintained at 800 rpm for 20 min at 22 °C to ensure proper mixing and coating. This approach ensured a uniform CHIT coating on the lipid vesicles, enhancing their stability and modifying the drug release profile [32].

### 2.3. Animals

Male Swiss white mice were used in the study, each weighing 20–25 g, provided by the National Medical-Military Institute for Research and Development in Băneasa, Bucharest, Romania, the Biobase at ‘Grigore T. Popa’ University of Medicine and Pharmacy and CEMEX, the Advanced Research and Development Center for Experimental Medicine in Iași, Romania. The animals were individually housed in Plexiglass cages and provided a 7-day acclimatization period under carefully controlled conditions. The laboratory environment maintained a temperature of 21 ± 2 °C, relative humidity between 50% and 70%, and a 12 h light/dark cycle. Standard granulated food and tap water were offered freely, and daily food consumption was recorded. The mice were observed regularly to monitor their overall behavior and well-being.

The mice were randomly assigned into groups of five animals and subjected to oral administration once daily using a polypropylene tube (size no. 16–20, length 5 cm). The treatment regimen for each group was as follows: control group: the animals were administered 0.1 mL of distilled water/10 g of body weight; IND group: the animals received a solution of IND at a dose of 5 mg/kg body weight; IND-ves group: the animals were treated with CHIT-based vesicles encapsulating IND, administered at a dose of 5 mg/kg body weight.

### 2.4. Ethical Aspects

The experimental research protocol was approved under Certificate No. 362/28.11.2023 (Project Authorization No. 69/15.01.2023) and conducted in accordance with the ethical guidelines established by the Committee for Research and Ethical Issues at ‘Grigore T. Popa’ University of Medicine and Pharmacy, Iași, Romania. All procedures adhered to international ethical standards for the care and use of laboratory animals, ensuring compliance with established regulations and best practices for animal research [48,49].

### 2.5. Biocompatibility Evaluation

#### 2.5.1. Blood Analysis

At two time points in the experiment (one day and seven days after the administration of the test substances), blood samples were collected from the lateral caudal vein [50] for hematological and biochemical analysis, which were performed using the 5 DIFF BF-5180 Analyzer (DIRUI, Changchun, China).

The following parameters were measured: serum hemoglobin levels, leukocyte formula (percentage values of neutrophilic polymorphonuclear cells-PMN, lymphocytes-L, monocytes-M, eosinophils-E, and basophils-B), blood levels of alanine aminotransferase (ALT), aspartate aminotransferase (AST), and lactate dehydrogenase (LDH), and serum levels of urea and creatinine.

As part of the evaluation of the immune defense, the following parameters were analyzed: opsonic capacity (OC), bactericidal capacity of peritoneal macrophages (BC), phagocytic capacity of PMN from peripheral blood (PC). Peritoneal macrophages were extracted through peritoneal lavage using Hanks solution, which was maintained at a constant temperature of 37 °C to ensure optimal collection conditions. After extracting the peritoneal fluid, the samples were centrifuged for 10 min at 1000 rpm to separate the cells. Following centrifugation, the macrophages were exposed to various cultures of *Staphylococcus aureus* 94, suspended in a 0.2% glucose broth solution diluted 1:1000 with saline. These samples were incubated at 37 °C for 48 h to allow bacterial growth. Subsequently, the macrophages were transferred to fresh culture media, and bacterial colony formation was evaluated by observation on culture plates. This method enabled the identification and quantification of bacterial colonies, contributing to the analysis of macrophage interactions with *Staphylococcus aureus* [51].

To examine the effect of the tested substances on oxidative processes, the activities of malondialdehyde (MDA), glutathione peroxidase (GPx), and superoxide dismutase (SOD) were measured using a Shimadzu Pharma Spec 1700 UV-Vis spectrophotometer (San Diego, CA, USA). Blood samples were drawn into vacutainer tubes containing EDTA, and after centrifuging at 1500 rpm for around 15 min, 1 mL of plasma was extracted. MDA levels were quantified using the thiobarbituric acid method with RANSOD kits from RANDOX Laboratories Ltd. (Crumlin, UK) [52]. The activity of GPx in plasma was determined by a reduction method involving nicotinamide adenine dinucleotide phosphate, using RANSEL kits from RANDOX Laboratories Ltd. [53]. Blood SOD activity was assessed through a colorimetric method with the RANSOD kit from RANDOX Laboratories Ltd., incorporating reagents such as xanthine and xanthine oxidase [54].

#### 2.5.2. Histopathological Examination

The animals were sacrificed and tissue samples from the liver, kidney, and stomach were collected for histopathological analysis. These tissues were fixed in a 10% formalin solution and then embedded in paraffin using a Leica TP 1020 rotary tissue processor (Leica Biosystems Nussloch GmbH, Nussloch, Germany). The samples were sectioned to a thickness of 5 μm using a microtome (Semi-Automatic Rotary Microtome CUT 5062-SLEE, Main, Germany). Staining was performed with Masson’s trichrome method (yellow, blue, and green), which helps differentiate collagen fibers from muscle tissue. The sections were examined using a Nikon TI Eclipse optical microscope (Tokyo, Japan), paired with a Nikon Coolpix 950 digital camera offering 3× optical zoom and a resolution of 1600 × 1200 (1.92 MP).

### 2.6. The In Vivo Kinetics Release Profile Assessment

The release pattern of IND from lipid vesicles was determined by measuring the drug concentration in the blood of mice at multiple time points. The animals were anesthetized using isoflurane at a concentration of 1% to reduce discomfort. Before administering IND-ves, 0.3 mL of blood (collected with heparin) was drawn to determine the baseline drug concentration, which was zero at time point 0. Following administration, further blood samples (with heparin) were collected at regular intervals (15 min, 30 min, 60 min, 90 min, 2 h, 3 h, 4 h, 5 h, 6 h, and 8 h after administration). The concentration of IND released into the bloodstream was quantified using a high-performance liquid chromatography (HPLC) system (Agilent 1100, Santa Clara, CA, USA). The system was equipped with an ultraviolet (UV) absorbance detector, set to a wavelength of 254 nm for precise IND detection. The mobile phase composition consists of a mixture of acetonitrile and phosphate buffer (pH 3.0) in a 60:40 (*v*/*v*) ratio. A 20 µL sample was injected into the HPLC system. The method ensured intra-day and inter-day precision, with relative standard deviation values remaining below 2%. The limit of detection was 0.05 µg/mL, and the limit of quantification was 0.15 µg/mL. The calibration curve range was 0.1–50 µg/mL, with an R^2^ value of 0.9765, confirming linearity.

### 2.7. The Nociceptive Reactivity Testing

The nociceptive response was evaluated using the tail-flick test, a model for somatic pain that measures sensitivity changes by applying thermal stimuli to the animal’s tail after administering the test substance [55]. The test recorded the time it took for the animal to respond when its tail was exposed to a heat source set at 52 °C (LE7106 Tail-flick Meter, PanLab Harvard Apparatus, Barcelona, Spain). The response latency was measured at various time points: before administration (baseline) and at 15, 30, 60, and 90 min, as well as 2, 4, 6, 8, and 10 h post-administration. To prevent heat-induced tail damage, the maximum exposure time (cut-off time) was set at 12 s. In this pain model, an increase in response time compared to baseline indicates an antinociceptive effect of the tested substances, suggesting reduced pain sensitivity. Conversely, a decrease in response latency signifies a hyperalgesic response, reflecting heightened pain sensitivity.

The intensity of the antinociceptive effect was quantified by converting latency measurements into a percentage of the maximum possible effect (% MPE), calculated as follows [56]:
% MPE = (tested latency − baseline latency) × 100/(cut-off time − baseline latency)


### 2.8. Data Analysis

The findings were reported as mean values accompanied by the standard deviation (S.D.) for each measurement time point. The data were processed and analyzed using SPSS software, version 24.0 for Windows and Statistics Toolbox in Matlab version 9.8.0.132.3502-2020a (The MathWorks, Inc., Natick, MA, USA). Appropriate statistical functions within the software were utilized to derive the necessary parameters for describing the data distribution. Statistical significance was determined for *p*-values below 0.05 and 0.01 when compared to the control group.

## 3. Results

In our previous research, we employed a molecular droplet self-assembly technique to develop lipid microvesicles encapsulating IND, made from phosphatidylcholine and stabilized with chitosan. These vesicles had an average size of 317.6 nm and a polydispersity index of 0.364. The systems displayed uniform morphology, high encapsulation efficiency, and enhanced stability, with a zeta potential of 24 mV. The microscopic analysis showed that the CHIT-coated vesicles maintained stability, in contrast to the aggregation observed in uncoated vesicles. FTIR analysis identified characteristic peaks associated with chitosan functional groups, confirming its presence on the vesicle surface. UV-Vis spectroscopy demonstrated an encapsulation efficiency of 85% for IND in the microvesicles [32].

The in vitro studies demonstrated a sustained release of IND from the lipid vesicles, compared to the immediate release of the free drug. The in vitro hemocompatibility evaluation showed only a slight level of hemolysis, suggesting that the microvesicles exhibit a favorable safety profile. This finding supports the potential of these microvesicles as effective and biocompatible delivery systems for incorporating IND [32].

### 3.1. Hematological and Biochemical Results

The in vivo research performed did not show any statistically significant changes in the erythrocyte count in animals treated with IND and IND-ves compared to those in the control group (Table 1). This lack of differences was consistent both 24 h and 7 days after administration of the tested substances, highlighting the stability of the hematological response in the animals with these treatments over the observed time periods.

The use of IND and IND-ves did not lead to substantial changes in the percentage of PMN compared to the control group throughout the experiment. The percentage of Ly remained similar in animals treated with IND and IND-ves, as well as those in the control group, at both 24 h and 7 days after administration. Additionally, no important variations in the percentage of E, M, B were detected between the animals treated with IND or IND-ves and those from the control group at both evaluation points, suggesting that the tested substances did not have a measurable impact on the mentioned cell populations.

The direct intragastric administration of IND and IND-ves did not result in notable changes in blood levels of ALT, AST, or LDH compared to the control group both at 24 h and 7 days after administration. Serum levels of urea and creatinine showed no considerable variations in animals treated with IND and IND-ves compared to the control group at both moments of testing in the experiment (Table 2).

The administration of IND and IND-ves did not have a significant impact on the OC of peritoneal macrophages one week after treatment compared to the control group. Furthermore, the BC of peritoneal macrophages showed no notable changes between the treated and control groups. Additionally, the PC of neutrophils in peripheral blood remained unchanged between tested animals and the control group 7 days post-administration (Table 3). These observations supports the idea that administration of these compounds did not disrupt the local immune defense mechanisms of the animals.

SOD activity showed no significant differences between the animals treated with IND and IND-ves versus the control group after 24 h, as well as after 7 days. Similarly, the serum level of GPx did not exhibit significant fluctuations in IND and IND-ves groups compared to the distilled water group at both time points assessed. Additionally, the serum levels of MDA, a marker of lipid peroxidation and oxidative stress, remained unmodified in animals treated with IND and IND-ves compared to the control group 1 day and 7 days post-administration (Table 4). These results indicate that the use of IND and IND-ves does not have a measurable impact on oxidative processes in mice, demonstrating good cellular tolerance and stability of the body’s natural antioxidant mechanisms in the presence of these substances in both short and medium terms.

### 3.2. Histological Aspects

The evaluation results indicated a normal liver morphology with a well-organized and clearly defined structure. The central vein was distinctly outlined, with hepatocytes uniformly arranged in layers around it. The hepatic sinusoids, the liver’s specialized blood vessels, were visible and appeared normal, without any signs of dilation or congestion. Additionally, the hepatocyte nuclei exhibited typical sizes and shapes, reflecting proper cellular function.

When compared to the control group that received distilled water, no significant structural changes were observed in the liver of mice treated with IND and IND-ves (Figure 1).

The histopathological study of kidney fragments from mice that received distilled water revealed a normal renal architecture, highlighting the integrity and functionality of this vital organ. In the control group mice, the glomeruli were observed to have a normal structure, with a well-defined basal membrane and an adequate number of mesangial cells. This suggests efficient renal function in blood filtration, which is essential for maintaining the body homeostasis. The proximal and distal convoluted tubules appeared healthy, with no signs of degeneration or damage. The tubular cells were evenly distributed, and their cytoplasm exhibited normal granularity, indicating proper metabolic activity. This is important for the efficient reabsorption of nutrients and water, contributing to urine formation. The use of IND and IND-ves did not lead to any notable structural alterations in the kidneys compared to the control group (Figure 2).

Comparative evaluation of the myocardium following administration of IND and IND-ves demonstrated the absence of significant changes in the structure of the cardiac muscle compared to the control group (Figure 3).

The histological analysis of stomach tissue samples collected from the control group mice revealed a normal and healthy structure. However, treatment with IND resulted in significant structural damage, including fragmented collagen fibers, reduced epithelial differentiation, and epithelial detachment, indicating harm to the gastric mucosa. Inflammatory responses were evident, with infiltration of mononuclear cells and vacuolization in the gastric gland cells, suggesting architectural damage. Microscopic examination of the stomach in animals treated with IND-ves showed no detectable structural changes, suggesting that the carrier system for IND preserved the gastric structure without visible alterations (Figure 4).

### 3.3. In Vivo Release Kinetics of IND from the IND-ves

In mice treated with non-loaded substance, IND levels increased rapidly and progressively, with a substantial rise observed between 15 and 120 min. The peak plasma concentration occurred at 30 min, followed by a sharp decrease over the next six hours. In contrast, IND-ves administration resulted in a delayed increase in plasma IND levels, beginning after a 90 min latency period and continuing for up to 6 h, with peak levels recorded at 3 h. Beyond this point, IND concentrations declined sharply, reaching minimal levels after eight hours. Comparing IND with IND-ves, it can be inferred that free IND rapidly reaches its peak plasma concentration, corresponding to its fast-acting but short-lived analgesic effect. In contrast, IND-ves exhibits a slower release profile, maintaining elevated blood concentrations for a longer period, which aligns with the observed analgesic effect between 2 and 6 h post-administration (Figure 5). Our analysis revealed that free IND exhibited a Cmax of 12.5 ± 0.8 µg/mL at tmax = 0.5 h, with a t1/2 of 1.8 ± 0.2 h and an AUC of 37.6 ± 2.5 µg·h/mL. In contrast, IND-ves demonstrated a slower release profile, with a lower Cmax occurring at tmax = 3.0 h, an extended t1/2, and a higher AUC (Table 5). These findings confirm that IND-ves provides a sustained release profile, prolonging drug presence in the bloodstream compared to free IND, which reaches its peak concentration more rapidly but declines at a faster rate.

### 3.4. The Somatic Nociceptive Reactivity

In the tail flick test, the treatment with free IND led to a rapid increase in reaction time, statistically significant compared to the control group, reaching a peak at 60 min. This elevated response was observed for up to four hours before gradually declining, returning to control levels after six hours. In contrast, the administration of IND-ves had no important impact on the response latency to the thermal stimulus applied to the tail during the first 90 min compared to the control group. However, after two hours, a significant progressive increase in response latency was observed, peaking at four hours and remaining elevated for up to eight hours in the experiment (Figure 6).

In order to assess the differences between the central values of the control and IND and of the control and IND-ves, we have used the Wilcoxon rank sum test rather than the *t*-test for independent variables. The latter relies on the normality assumption of the compared variables, which is not met here. The size of the dataset is too small for the normality assumption to be valid. The Wilcoxon rank sum test (also called the Mann–Whitney U test) is a non-parametric version of the two-sample *t*-test for independent samples, meaning that it does not rely on this assumption. This test is used to compare the median values of two groups, and it only requires that the observations from both groups are of a continuous type and independent of each other. The underlying null hypothesis of this test is that there is no difference between the compared groups. As the name of the test suggests, instead of using the raw data values to calculate the test statistic, it uses the ranks of the data.

Based on the Wilcoxon rank sum test, we have found significant differences between some of the median values of the control and IND and of the control and IND-ves in different time frames. This fact can also be observed in Figure 6, where ‘**’ means significantly different median values from control at 
α=0.01.


We utilized the percentage of maximum possible effect (%MPE) to measure the antinociceptive intensity, ensuring the reliability and validation of our results. IND treatment produced a significant rise in %MPE within the first 90 min, peaking at 60 min (%MPE = 43.6 ± 0.23) in the tail-flick test. Meanwhile, IND-ves administration extended the duration of antinociceptive effects, showing a notable increase between 2 and 10 h, with the highest %MPE observed at 4 h (%MPE = 43.2 ± 0.28) (Figure 7).

Supplementarily, we analyzed the correlation between the plasma concentration of IND released from IND-ves and the latency response in the tail-flick test in mice. Figure 7 illustrates the regression line approximation between these two variables. Our findings indicate a strong positive linear correlation between the plasma IND concentration from IND-ves and the latency response following IND-ves administration in the tail-flick test. The Pearson correlation coefficient (r = 0.9226) is statistically significant at a significance level of 0.001 (*p* = 0.000143, 95%CI: [0.70, 0.98]).

Additionally, the coefficient of determination (R^2^ = 0.851) suggests that approximately 85% of the variability in latency response time can be attributed to the plasma concentration of IND-ves in vivo and vice versa (Figure 8).

## 4. Discussions

Numerous studies have explored different strategies for developing nanosystems to encapsulate IND, aiming to enhance its solubility, stability, bioavailability, and controlled release properties. These advancements highlight the diverse approaches available for optimizing IND’s therapeutic potential [57,58].

One widely investigated method involves lipid-based nanocarriers, incorporating phospholipids and lipid emulsions to improve IND solubility and bioavailability. These formulations often utilize combinations of lipids, surfactants, and cosolvents to create stable delivery systems [59,60]. Liposomal structures, particularly those composed of multilayered vesicles, have been developed using components such as stearylamines, cholesterol, and chitosan (CHIT) to facilitate IND stabilization and controlled release. Furthermore, sterically stabilized liposomes have demonstrated enhanced stability, ensuring prolonged therapeutic efficacy [61].

CHIT has gained recognition as a promising material for IND-loaded nanosystems due to its biocompatibility, biodegradability, antibacterial properties, and mechanical strength [62]. Ionotropic gelation techniques have been employed to fabricate CHIT-based nanoparticles, utilizing tripolyphosphate anions in aqueous solutions to produce IND-ves nanoparticles with enhanced stability and prolonged drug release. Meanwhile, supercritical fluid technology has been employed to process IND using supercritical carbon dioxide under high-pressure conditions, leading to the formation of polyethylene glycol (PEG)-stabilized nanoparticles with precisely controlled particle size, high purity, and minimal solvent usage [63].

Another approach involves modifying the release profile of IND through triacylglyceride-based conjugates, which integrate phosphatidylcholine, palmitic acid, or stearic acid to improve lipid compatibility and drug retention [64]. In vivo experiments have shown that IND-loaded lipid vesicles stabilized with CHIT extend analgesic effects significantly longer than unencapsulated IND, enabling a gradual and sustained release [65].

Polymeric nanosystems, particularly those incorporating biodegradable and biocompatible materials like polycaprolactone (PCL) [66] and poly(lactic-co-glycolic acid) (PLGA) [67], have also been extensively studied for IND encapsulation. Such systems utilize solvent evaporation, nanoprecipitation, or double emulsion techniques to ensure controlled and extended drug release [68,69].

Additionally, alternative carrier systems have been developed to modify IND release, including copolymeric networks comprising poly(2-hydroxyethyl methacrylate-co-3,9-divinyl-2,4,8,10-tetraoxaspiro[5.5]-undecane) or poly(aspartic acid) as stabilizing colloids. Other polymerization-based nanosystems incorporated hyaluronic acid and poly(itaconic anhydride-co-3,9-divinyl-2,4,8,10-tetraoxaspiro[5.5]-undecane) derivatives for controlled release applications [70,71].

To minimize gastric irritation and ensure intestinal release, gastroresistant nanoparticles incorporating Eudragit L100, PEG, and polysorbate 80 have been designed [72,73]. Furthermore, amorphous nanosuspensions produced via aqueous wet milling have integrated IND with polyvinylpyrrolidone, ensuring stability in solution while preserving bioavailability [74].

Various techniques such as spray drying, coprecipitation, and solvent evaporation have been applied to enhance dissolution and stability, often incorporating cyclodextrins for improved bioavailability and prolonged analgesic effects in pre-clinical models [75,76,77].

A novel approach, electrohydrodynamic atomization, has also been explored for producing uniform and monodisperse IND nanoparticles. By applying a high voltage to an IND solution containing PEG, this technique facilitates controlled and targeted drug delivery [78].

Wang and co. developed IND-functionalized linear polyglycidol micelles which showed strong surface activity and superior antitumor effects, demonstrating preferential uptake by COX-2-positive cancer cells, highlighting their potential for targeted therapy [79].

Other researchers developed IND-loaded glycosylated nanostructured lipid carriers to selectively target macrophages in osteoarthritis, reprogramming them to an anti-inflammatory phenotype and offering a promising strategy for managing inflammation in this pathological state [80].

The method employed in our study proved to be efficient in obtaining vesicles that are uniform in size, have a modified release in vitro model, and are moderately stable. Furthermore, we tested the biochemical and biocompatibility parameters and in vivo kinetics of IND-ves.

Laboratory analyses confirmed that the administration of both IND and IND-ves did not lead to significant alterations in erythrocyte count, indicating that these treatments did not negatively affect red blood cell populations or disrupt hematological stability. This consistency was maintained in both short-term (24 h) and long-term (7 days) evaluations. Similarly, the proportions of PMNs, Ly, E, M, and B remained stable in treated animals, suggesting that IND and IND-ves did not impact immune cell composition or provoke an immune response. Serum analysis revealed no significant changes in liver enzyme levels, indicating that neither IND nor IND-ves compromised the liver function, thus supporting their favorable hepatic tolerability. Likewise, serum levels of urea and creatinine remained stable, confirming that kidney function was not impaired, ensuring renal tolerance under the evaluated conditions and time intervals.

Markers of oxidative stress, such as SOD, GPx activity, and MDA levels, showed no significant changes. This stability suggests that the oxidative function of immune cells remained intact, confirming that neither IND nor IND-ves induced oxidative stress or activated non-specific immune responses. Furthermore, the bactericidal and phagocytic capacities of peritoneal macrophages and neutrophils—key immune defense cells—remained unaffected by the treatment. The preservation of BC indicates that neither IND, CHIT, nor IND-ves altered macrophage function, while the maintenance of PC suggests that neutrophils retained their ability to recognize, engulf, and digest pathogens, ensuring effective innate immune responses. These findings collectively demonstrate that IND and IND-ves did not disrupt essential immune cell functions, were well-tolerated, and did not elicit proinflammatory or allergic responses. The treatments did not cause notable shifts in hematological, biochemical, or immunological parameters, further reinforcing their biocompatibility.

The microscopic evaluation of treated tissues, including the liver, kidney, muscle fibers, and myocardium, revealed normal morphology across all groups, suggesting that IND and IND-ves did not cause structural damage to these organs. Liver architecture showed no alterations, indicating no compromise to liver integrity. Similarly, kidney architecture remained intact in animals administered IND and IND-ves, confirming that the lipid vesicle formulation does not negatively affect renal structure and may serve as a promising drug delivery system. The integrity of the myocardium, including coronary arteries and capillaries, remained unaffected, ensuring cardiac circulation was not compromised. In contrast, gastric mucosa integrity was maintained in the control and IND-ves groups, with a well-preserved epithelial layer and normal gastric gland structure, indicating proper gastric function. However, in the IND-treated group, the histological analysis revealed fragmented collagen fibers, reduced epithelial differentiation, and structural disorganization, which indicated damage to the extracellular matrix and gastric mucosa. The gastric mucosa was diffusely thickened with signs of epithelial detachment, pointing to acute lesions. Vascular congestion and mononuclear cell infiltration in the submucosa reflected an inflammatory response. Vacuolization in gastric gland cells suggested potential functional impairment. In animals treated with IND-ves, no significant gastric architectural changes were observed, indicating that the lipid-vesicle-based formulation may offer protection against IND-induced gastric damage, preserving mucosal structure and function.

Chitosan is known to have mucoadhesive properties, which enhance its ability to adhere to the gastric mucosa and form a protective layer [81]. This adhesion prolongs the retention time of chitosan at the gastric site, reducing direct exposure of the mucosa to irritants and acidic conditions [82,83]. Additionally, chitosan has been reported to enhance mucus secretion, which can further strengthen the mucosal barrier against gastric damage [84].

Moreover, chitosan exhibits bioadhesive interactions with mucins due to its cationic nature, forming electrostatic bonds with the negatively charged sialic acid and sulfate residues of mucosal glycoproteins [82]. This contributes to the stabilization of the gastric mucosa and may aid in tissue regeneration. Furthermore, its ability to chelate bile acids and suppress pepsin activity provides an additional protective mechanism by reducing enzymatic degradation of the mucus layer [85]. These combined properties of chitosan, mucoadhesion, enhancement of mucus secretion, electrostatic stabilization, and enzymatic inhibition, are key factors contributing to its gastric-protective effects.

The in vivo release kinetics indicates a sustained release pattern, highlighting the effectiveness of the IND-ves system in achieving a prolonged release profile.

Oral administration of IND-ves induced notable antinociceptive effects emerging after 2 h, reaching peak intensity between 4 and 6 h, and persisting for up to 8 h. The delayed onset of the analgesic effect in the tail-flick test can be attributed to the gradual release of IND from the lipid vesicle structure, resulting in slower absorption and a delay in reaching the effective plasma concentration required for pain relief. This characteristic is typical of extended release formulations, which sustain therapeutic effects for a prolonged period but take longer to initiate analgesia.

The latency period in response occurs because encapsulated IND must first be released gradually from the phospholipid shell, cross biological barriers, and be absorbed into blood stream. This process is influenced by the physicochemical properties of the vesicles, their stability in biological environments, and transport mechanisms across cell membranes. Additionally, the diffusion of IND to target tissues may be affected by its interactions with plasma proteins and hepatic metabolism. These factors explain why the analgesic effect of IND-ves takes longer to manifest compared to free IND but is sustained for a longer duration.

The correlation between the blood levels of IND released from IND-ves and the latency reactivity in this somatic pain model indicates that changes in reaction time to thermal noxious tail stimulation are closely correlated with variations in plasma IND concentration.

## 5. Conclusions

Both IND and IND-ves exhibited excellent biocompatibility, with no significant impact on hematological, biochemical, or immunological parameters. The lipid vesicle formulation offered additional advantages, including protection against drug degradation, enhanced bioavailability, and sustained release. IND-ves administration showed no adverse effects on liver and kidney function or oxidative stress markers, confirming its safety profile. Histopathological analysis further demonstrated that lipid vesicles shield vital organs, such as the stomach, from drug-induced damage.

The sustained release properties of IND-ves facilitated gradual drug release, leading to prolonged analgesic effects in the tail-flick test in mice. These findings highlight the potential of IND-loaded lipid vesicles to enhance therapeutic efficacy while reducing side effects. Further studies are needed to fully explore their clinical applications.

## Figures and Tables

**Figure 1 pharmaceutics-17-00523-f001:**
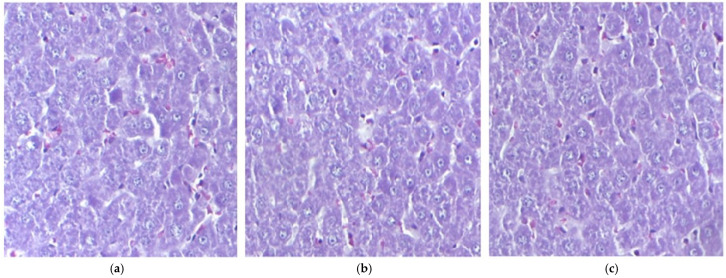
The histological liver structure in mice belonging to the following groups: Control (**a**), IND (**b**), IND-ves (**c**). Masson trichrome stain ×20, scale bar = 100 µm.

**Figure 2 pharmaceutics-17-00523-f002:**
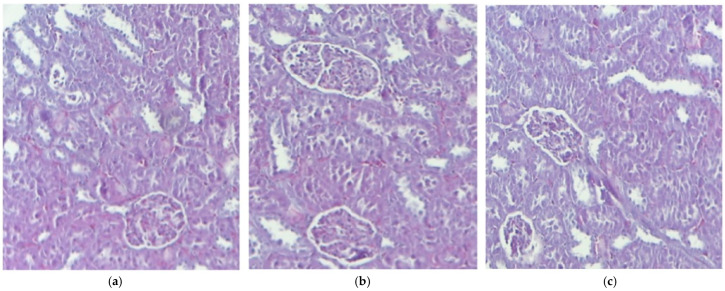
The histological kidney structure in mice belonging to the following groups: Control (**a**), IND (**b**), IND-ves (**c**). Masson trichrome stain ×20, scale bar = 100 µm.

**Figure 3 pharmaceutics-17-00523-f003:**
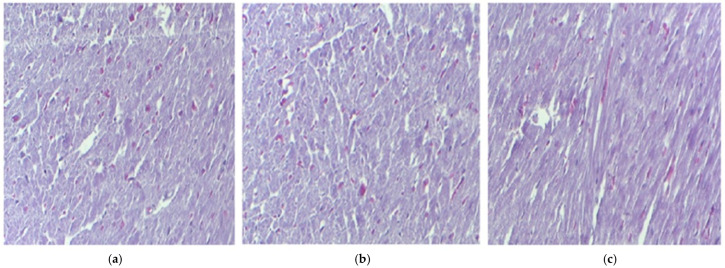
The histological myocardium structure in mice belonging to the following groups: Control (**a**), IND (**b**), IND-ves (**c**). Masson trichrome stain ×20, scale bar = 100 µm.

**Figure 4 pharmaceutics-17-00523-f004:**
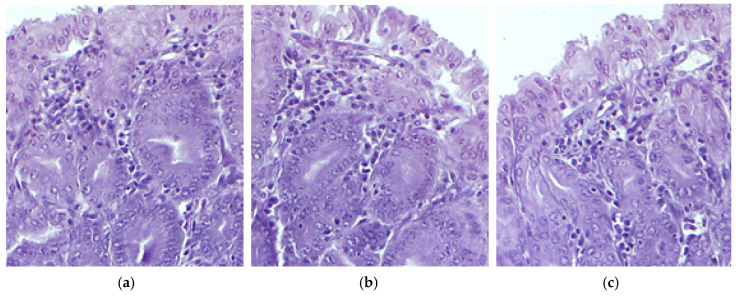
The histological stomach structure in mice belonging to the following groups: Control (**a**), IND (**b**), IND-ves (**c**). Masson trichrome stain ×20, scale bar = 100 µm.

**Figure 5 pharmaceutics-17-00523-f005:**
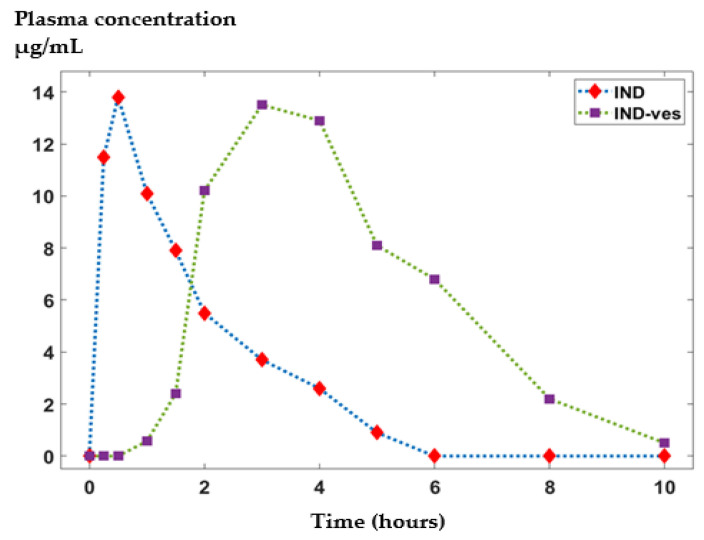
The in vivo release kinetics of IND from IND-ves compared to free IND.

**Figure 6 pharmaceutics-17-00523-f006:**
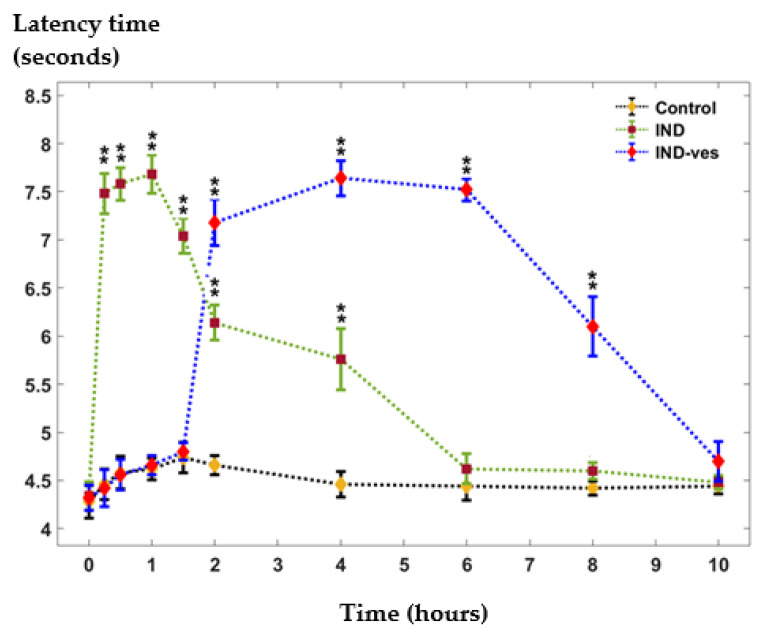
The effects of IND-ves in tail-flick test. Data are expressed as mean ± S.D. of mean for five mice in a group. ** *p* < 0.01 significant versus control group.

**Figure 7 pharmaceutics-17-00523-f007:**
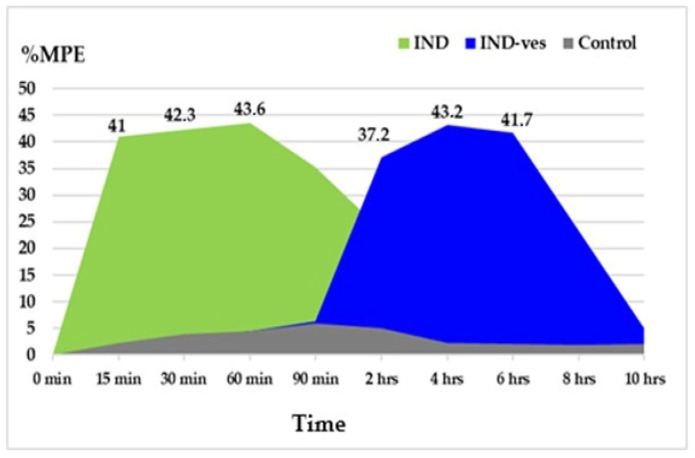
Quantification of the analgesic effect intensity of IND-ves in the tail-flick test.

**Figure 8 pharmaceutics-17-00523-f008:**
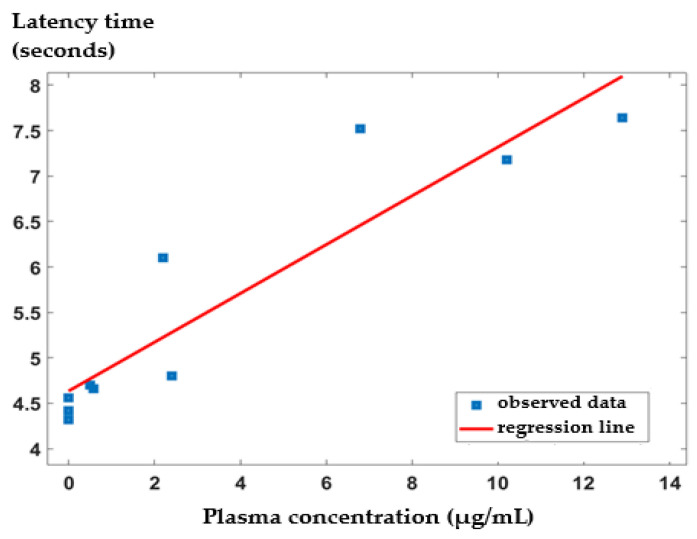
The correlation between the plasma concentration of IND-ves in vivo and the latency time response of IND-ves in the tail-flick test.

**Table 1 pharmaceutics-17-00523-t001:** The number of erythrocytes and the percentage of leukocyte formula elements. PMN—neutrophilic polymorphonuclear cells, Ly—lymphocytes, M—monocytes, E—eosinophils, and B—basophils. Values are expressed as mean ± standard deviation (SD) for five mice in a group.

	Evaluation Points	GR (mil/μL)	Leukocyte Formula Elements (%)				
			PMN	Ly	E	M	B
Control	24 h	8.2 ± 1.3	28.4 ± 1.9	65.9 ± 5.9	0.1 ± 0.03	5.4 ± 0.3	0.2 ± 0.01
	7 days	8.4 ± 1.1	29.1 ± 1.5	65.2 ± 5.5	0.2 ± 0.01	5.3 ± 0.1	0.2 ± 0.01
IND	24 h	8.3 ± 0.5	28.9 ± 2.3	65.4 ± 6.3	0.1 ± 0.01	5.4 ± 0.5	0.2 ± 0.01
	7 days	8.6 ± 1.1	29.3 ± 2.7	65,1 ± 5.9	0.1 ± 0.01	5.3 ± 0.3	0.2 ± 0.01
IND-ves	24 h	8.4 ± 0.9	28.6 ± 2.3	65.8 ± 6.3	0.1 ± 0.03	5.5 ± 0.3	0.2 ± 0.01
	7 days	8.7 ± 0.3	28.5 ± 2.5	65.5 ± 5.7	0.2 ± 0.03	5.6 ± 0.1	0.2 ± 0.01

**Table 2 pharmaceutics-17-00523-t002:** The serum levels of the assessment parameters for liver and kidney function. Values are expressed as mean ± standard deviation (SD) for five mice in a group.

	Evaluation Points	ALT (U/mL)	AST (U/mL)	LDH (U/L)	Urea (mg/dL)	Creatinine (mg/dL)
Control	24 h	39.5 ± 3.9	153.7 ± 12.7	322.3 ± 19.7	27.1 ± 3.1	0.4 ± 0.03
	7 days	39.3 ± 4.5	156.1 ± 14.3	325.5 ± 18.9	27.5 ± 3.3	0.5 ± 0.01
IND	24 h	38.3 ± 5.1	155.7 ± 13.5	325.5 ± 19.3	26.5 ± 2.9	0.5 ± 0.01
	7 days	38.5 ± 4.7	157.5 ± 13.9	326.9 ± 19.5	26.3 ± 3.7	0.4 ± 0.03
IND-ves	24 h	38.7 ± 4.3	155.9 ± 14.5	323.3 ± 18.7	27.3 ± 2.5	0.4 ± 0.03
	7 days	39.1 ± 4.9	158.1 ± 14.3	327.7 ± 20.3	27.7 ± 2.9	0.4 ± 0.01

**Table 3 pharmaceutics-17-00523-t003:** The serum levels of OC, BC, and PC. Values are expressed as mean ± standard deviation (SD) for five mice in a group.

	Evaluation Points	OC (Colonies/mL)	BC (Colonies/mL)	PC (Colonies/mL)
Control	7 days	757.9 ± 31.7	705.7 ± 30.5	525.5 ± 20.9
IND	7 days	760.1 ± 34.5	708.8 ± 31.3	528.5 ± 23.3
IND-ves	7 days	759.5 ± 33.7	706.1 ± 30.7	523.3 ± 24.1

**Table 4 pharmaceutics-17-00523-t004:** The serum levels of enzymes involved in oxidative processes. Values are expressed as mean ± standard deviation (SD) for five mice in a group.

	Evaluation Points	SOD (U/mg Protein)	GPx (µm/mg Protein)	MDA (nMol/mg Protein)
Control	24 h	27.5 ± 3.3	0.4 ± 0.01	16.7 ± 2.1
	7 days	27.7 ± 2.9	0.5 ± 0.03	16.9 ± 1.5
IND	24 h	28.3 ± 2.9	0.5 ± 0.03	17.1 ± 1.7
	7 days	28.1 ± 3.1	0.6 ± 0.01	17.3 ± 2.3
IND-ves	24 h	27.5 ± 3.3	0.4 ± 0.01	17.5 ± 1.5
	7 days	28.3 ± 3.1	0.5 ± 0.01	16.7 ± 2.1

**Table 5 pharmaceutics-17-00523-t005:** In vivo pharmacokinetic parameters of IND compared to IND-ves.

	$C_{max}$ (µg/mL)	$t_{max}$ (hours)	$t_{1/2}$ (hours)	AUC (µg·h/mL)
Free IND	13.8	0.5	1.9	28.375
IND-ves	13.5	3	4.5	58.735

*C_max_*, maximum concentration; *t_max_*, time to maximum concentration, *t*_1/2_ elimination half-life; AUC_0-last_, area under the curve from time of administration up to the time of the last quantifiable concentration.

## Data Availability

Data are contained within the article.

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
