# Peer review of "Chitosan-Stabilized Lipid Vesicles with Indomethacin for Modified Release with Prolonged Analgesic Effect: Biocompatibility, Pharmacokinetics and Organ Protection Efficacy"

_pharmaceutics, 2025, doi:10.3390/pharmaceutics17040523_

Round 1
Reviewer 1 Report
Comments and Suggestions for Authors
General comments
The manuscript entitled “Chitosan-Stabilized Lipid Vesicles With Indomethacin For Modified Release With Prolonged Analgesic Effect: Biocompatibility, Pharmacokinetics And Organ Protection Efficacy” deals with in vivo experiments on mice that were performed to evaluate pharmacokinetic and biocompatibility parameters after oral administration of indomethacin and indomethacin loaded lipid vesicles coated with chitosan.
Although several papers report the use of nanocarriers to limit the side-effects and to improve the bioavailability of indomethacin after its oral administration, this manuscript describes an in vivo investigation on animals whose results could provide useful information about the use of nanocarriers for oral administration of anti-inflammatory drugs.
The authors carried out several in vivo determinations but some experimental procedures should be better described.
Specific comments
Line 29. The meaning of the abbreviation CHIT should be specified.
Line 113, 508 and 511. References should be reported using numbers in brackets. Please, correct.
Line 131. The aim of the work is poorly described. Please, provide a more accurate description of the aim of the work.
Line 149. Although the authors have already published a paper on chitosan-coated lipid vesicles for indomethacin delivery, the amount of each ingredient used to prepare lipid vesicles should be reported.
Line 178. The authors state “the animals received a solution of IND at a dose of 5 mg/kg body weight”. Indomethacin is a poorly water-soluble drug. Which solvent did the authors use to prepare a solution of IND?
Line 250. The authors performed HPLC analyses but they did not report the experimental conditions used. Please, report the following parameters: mobile phase composition, injected volume, accuracy and precision of the method, limit of quantification and limit of detection, range of the calibration curve.
Line 284. Microvesicles should read nanovesicles. Please, correct.
Table 1. Tables should be self-explanatory. Therefore, the meaning of the abbreviations (PMN, Ly, etc.) should be reported in a legend. In addition, in Table 1,2,4 the text of the manuscript is embedded into the Table. Please, correct.
In section “In Vivo Release Kinetics of IND From the IND-ves”, for a better comprehension of the results, the authors should calculate and report in a Table the pharmacokinetic parameters (Cmax, tmax, t1/2, AUC) of IND and IND-ves.
Author Response
Distinguished Reviewer,
Thank you for the time on reviewing our manuscript and for the pertinent suggestions that helped us to improve the content of the article. We have addressed point-by-point all the comments and have made corrections in our manuscript using the track changes tool.
General comments
The manuscript entitled “Chitosan-Stabilized Lipid Vesicles With Indomethacin For Modified Release With Prolonged Analgesic Effect: Biocompatibility, Pharmacokinetics And Organ Protection Efficacy” deals with in vivo experiments on mice that were performed to evaluate pharmacokinetic and biocompatibility parameters after oral administration of indomethacin and indomethacin loaded lipid vesicles coated with chitosan.
Although several papers report the use of nanocarriers to limit the side-effects and to improve the bioavailability of indomethacin after its oral administration, this manuscript describes an in vivo investigation on animals whose results could provide useful information about the use of nanocarriers for oral administration of anti-inflammatory drugs.
The authors carried out several in vivo determinations but some experimental procedures should be better described.
Thank you for your comments and for highlighting the novelty of our in vivo investigation. We agree that while several studies have explored the use of nanocarriers to enhance the bioavailability and reduce the side effects of indomethacin, our research contributes valuable insights into their practical application for oral administration of anti-inflammatory drugs in animal models. We believe these findings could further inform the development of safer and more effective drug delivery systems.
Specific comments
Line 29. The meaning of the abbreviation CHIT should be specified.
Thank you for this observation. We have specified in the text the meaning of the abbreviation for CHIT.
Line 113, 508 and 511. References should be reported using numbers in brackets. Please, correct.
Thank you for these observations. We have corrected and included these authors in the references section.
Line 131. The aim of the work is poorly described. Please, provide a more accurate description of the aim of the work.
Thank you for your valuable feedback regarding the description of the aim of our work. We acknowledge that the initial statement may not have been sufficiently precise. To address this, we have revised the aim to clearly reflect the objectives of our study.
The objective of this study was to evaluate the biocompatibility, release kinetics, and analgesic potential of indomethacin-loaded chitosan-stabilized lipid vesicles (IND-ves) in comparison to free IND. Specifically, we sought to determine the pharmacokinetic advantages of IND-ves, assess its impact on somatic nociceptive reactivity in mice, and investigate its effects on physiological parameters such as liver and kidney function, oxidative stress markers, and immune cell activity. Through this comprehensive evaluation, our aim was to establish whether the IND-ves formulation could offer a more effective and safer alternative for NSAID therapy, particularly by enhancing drug release, prolonging analgesic efficacy, and minimizing gastrointestinal irritation.
Line 149. Although the authors have already published a paper on chitosan-coated lipid vesicles for indomethacin delivery, the amount of each ingredient used to prepare lipid vesicles should be reported.
We appreciate the reviewer’s comment regarding the need to specify the amount of each ingredient used in the preparation of lipid vesicles. We have inserted, in the manuscript, the amount of each ingredient used to prepare lipid vesicles.
Line 178. The authors state “the animals received a solution of IND at a dose of 5 mg/kg body weight”. Indomethacin is a poorly water-soluble drug. Which solvent did the authors use to prepare a solution of IND?
We appreciate the reviewer’s observation regarding the solubility of IND and the need to specify the solvent used for its preparation. This aspect has already been mentioned in the manuscript, stating that ethanol was used as the solvent for IND. Ethanol was chosen due to its ability to dissolve IND efficiently while ensuring homogeneity of the solution before administration.
Line 250. The authors performed HPLC analyses but they did not report the experimental conditions used. Please, report the following parameters: mobile phase composition, injected volume, accuracy and precision of the method, limit of quantification and limit of detection, range of the calibration curve.
Thank you for your observation regarding the reporting of HPLC conditions. We acknowledge the importance of providing a comprehensive description of the method and have now included, in the manuscript, the detailed experimental conditions for the HPLC analysis as follows:
The mobile phase composition consists of a mixture of acetonitrile and phosphate buffer (pH 3.0) in a 60:40 (v/v) ratio. A 20 µL sample was injected into the HPLC system. The method ensured intra-day and inter-day precision, with relative standard deviation values remaining below 2%. The limit of detection was 0.05 µg/mL, and the limit of quantification was 0.15 µg/mL. The calibration curve range was 0.1–50 µg/mL, with a R² value above 0.9765, confirming linearity.
Line 284. Microvesicles should read nanovesicles. Please, correct.
We appreciate your attention to the terminology used in our manuscript. However, we respectfully maintain the use of "microvesicles" rather than "nanovesicles" for our formulation. According to standard definitions in nanotechnology, nanovesicles typically refer to vesicles ranging in size between 1-100 nm. In contrast, our study reports vesicles with an average size of 317.6 nm, which falls outside the nanoscale range. Therefore, the term "microvesicles" more accurately describes our system. Based on this classification, we believe that the current terminology is appropriate and does not require modification. We hope this clarification adequately addresses the your concern.
Table 1. Tables should be self-explanatory. Therefore, the meaning of the abbreviations (PMN, Ly, etc.) should be reported in a legend. In addition, in Table 1,2,4 the text of the manuscript is embedded into the Table. Please, correct.
Thank you for your valuable feedback. We have inserted the meaning of the abbreviations.
You are absolutely right, there was text embedded in Tables 1, 2, and 4. We have fixed the issue accordingly.
In section “In Vivo Release Kinetics of IND From the IND-ves”, for a better comprehension of the results, the authors should calculate and report in a Table the pharmacokinetic parameters (Cmax, tmax, t1/2, AUC) of IND and IND-ves.
Thank you for the recommendation to include pharmacokinetic parameters for better comprehension of the results. In response, we have incorporated, in the manuscript, the key pharmacokinetic parameters, including maximum plasma concentration (Cmax), time to reach Cmax (tmax), elimination half-life (t1/2), and area under the curve (AUC), for both IND and IND-ves. AUC was calculated using trapezoid method, a numerical technique which assesses the area under the curve by dividing the total interval into smaller subintervals and calculating the area of each subinterval as a trapezoid. The area of each trapezoid is determined using the average of the function values at the endpoints of the subintervals, multiplied by the width of the subinterval, and then summing these areas provides an estimate of the total area under the curve.
According to your suggestion, we have summarized the pharmacokinetic in vivo parameters of IND and IND-ves in a table (inserted in the article), highlighting the key differences in the release profiles and plasma concentrations over time. These parameters further emphasize the proonged drug presence in the blood stream of IND-ves, compared to the faster onset of action and shorter duration for free IND.
Table. In vivo pharmacokinetic parameters of IND and IND-ves
|
|
(µg/mL) |
(hours) |
(hours) |
AUC (µg·h/mL) |
|
free IND |
13.8 |
0.5 |
1.9 |
28.375 |
|
IND-ves |
13.5 |
3 |
4.5 |
58.735 |
, maximum concentration; , time to maximum concentration, , elimination half-life; AUC0-last, area under the curve from time of administration up to the time of the last quantifiable concentration;
Calculation of AUC values for in vivo release kinetics
|
IND |
|||||
|
Time (hours) |
Concentration |
C1 + C2 |
Δ t |
AUC linear |
AUC total |
|
0 |
0 |
|
|
|
|
|
0.25 |
11.5 |
11.5 |
0.25 |
1.4375 |
1.4375 |
|
0.5 |
13.8 |
25.3 |
0.25 |
3.1625 |
4.6 |
|
1 |
10.1 |
23.9 |
0.5 |
5.975 |
10.575 |
|
1.5 |
7.9 |
18 |
0.5 |
4.5 |
15.075 |
|
2 |
5.5 |
13.4 |
0.5 |
3.35 |
18.425 |
|
3 |
3.7 |
9.2 |
1 |
4.6 |
23.025 |
|
4 |
2.6 |
6.3 |
1 |
3.15 |
26.175 |
|
5 |
0.9 |
3.5 |
1 |
1.75 |
27.925 |
|
6 |
0 |
0.9 |
1 |
0.45 |
28.375 |
|
8 |
0 |
0 |
2 |
0 |
28.375 |
|
10 |
0 |
0 |
2 |
0 |
28.375 |
|
IND-ves |
|||||
|
Time (hours) |
Concentration |
C1 + C2 |
Δ t |
AUC linear |
AUC total |
|
0 |
0 |
|
|
|
|
|
0.25 |
0 |
0 |
0.25 |
0 |
0 |
|
0.5 |
0 |
0 |
0.25 |
0 |
0 |
|
1 |
0.57 |
0.57 |
0.5 |
0.1425 |
0.1425 |
|
1.5 |
2.4 |
2.97 |
0.5 |
0.7425 |
0.885 |
|
2 |
10.2 |
12.6 |
0.5 |
3.15 |
4.035 |
|
3 |
13.5 |
23.7 |
1 |
11.85 |
15.885 |
|
4 |
12.9 |
26.4 |
1 |
13.2 |
29.085 |
|
5 |
8.1 |
21 |
1 |
10.5 |
39.585 |
|
6 |
6.8 |
14.9 |
1 |
7.45 |
47.035 |
|
8 |
2.2 |
9 |
2 |
9 |
56.035 |
|
10 |
0.5 |
2.7 |
2 |
2.7 |
58.735 |

Reviewer 2 Report
Comments and Suggestions for Authors
The manuscript presents an innovative study on chitosan-stabilized lipid vesicles (IND-ves) as a modified release system for indomethacin (IND), focusing on biocompatibility, pharmacokinetics, and organ protection efficacy. The manuscript is acceptable with significant revisions.
While statistical tests (e.g., ANOVA, post-hoc Tukey, Newman-Keuls tests) are mentioned, some sections lack clear statistical reporting:
Effect sizes should be included to assess clinical relevance.
Confidence intervals should be reported alongside p-values.
Normality assumptions for statistical tests should be verified and stated.
While the tail-flick test is an established model, additional pain models (e.g., formalin test or mechanical nociception) would strengthen the analgesic evaluation.
Some graphs lack detailed axis labeling and confidence intervals.
On line 283 present reference for your research.
on line 287 the zeta potential must be represented with a positive or negative value.
The manuscript does not specify important parameters such as:
Sonication amplitude and duration (e.g., “ultrasonic homogenizer used under controlled amplitude and temperature” is too vague).
Magnetic stirring speed (rpm) and duration for chitosan coating.
Temperature of ethanol evaporation (if applicable).
Lipid-to-drug ratio and exact concentration of IND in the vesicles.
How was chitosan coating efficiency confirmed?
Was the coating uniform or patchy?
Report EE% and LE% using HPLC or UV-Vis quantification.
Provide a calculation formula for encapsulation efficiency.
Discuss mechanistic reasons for gastric protection (e.g., chitosan mucoadhesive properties).
Several key nanoparticle characterization techniques are missing:
Transmission Electron Microscopy (TEM) or Scanning Electron Microscopy (SEM)
No confirmation of chitosan interaction with lipid vesicles- FTIR spectra would confirm chitosan-polymer bonding and IND encapsulation efficiency.
No assessment of IND crystallinity or polymorphic changes.XRD/DSC data would confirm whether IND remains amorphous or crystalline within the vesicles.
Author Response
Distinguished Reviewer,
We sincerely appreciate the time and effort you invested in reviewing our manuscript. Your insightful feedback has been incredibly helpful in enhancing and refining our work while providing essential direction. We have carefully considered each of your comments and made the necessary revisions, which are marked in red in the updated version. Below, we offer detailed responses to each point raised, addressing them individually.
The manuscript presents an innovative study on chitosan-stabilized lipid vesicles (IND-ves) as a modified release system for indomethacin (IND), focusing on biocompatibility, pharmacokinetics, and organ protection efficacy. The manuscript is acceptable with significant revisions.
Thank you for your positive feedback and recognition of our study. We truly appreciate your insightful comments and are glad that you find our work on chitosan-stabilized lipid vesicles valuable. Your acknowledgment of our focus on biocompatibility, pharmacokinetics, and organ protection efficacy is highly encouraging.
While statistical tests (e.g., ANOVA, post-hoc Tukey, Newman-Keuls tests) are mentioned, some sections lack clear statistical reporting:
Effect sizes should be included to assess clinical relevance.
Some graphs lack detailed axis labeling and confidence intervals.
Confidence intervals should be reported alongside p-values.
Thank you for this observation.
The effect sizes were assessed using the Statistics Toolbox in Matlab, version 9.8.0.1323502 (The MathWorks, Inc., Natick, Massachusetts, United States). All these statistical parameters were inserted in the manuscript.
Table. X Confidence intervals (CI - level of confidence 0.05) and Cohen’s d scores for the difference in medians of free IND, respectively of IND-ves and Control, in the tail flick test.
|
|
IND |
IND-ves |
||
|
Time (hours) |
95% CI |
Cohen’s d |
95% CI |
Cohen’s d |
|
0 |
[-0.40, 0.50] |
0.3328 |
[-0.30, 0.50] |
0.2329 |
|
0.25 |
[2.60, 3.60] |
14.2364 |
[-0.40, 0.50] |
0.2000 |
|
0.5 |
[2.60, 3.40] |
15.5963 |
[-0.40, 0.40] |
0.1069 |
|
1 |
[2.70, 3.40] |
16.4745 |
[-0.20, 0.30] |
0.3266 |
|
1.5 |
[1.90, 2.70] |
11.7988 |
[-0.20, 0.40] |
0.4092 |
|
2 |
[1.20, 1.80] |
8.8447 |
[2.10, 2.90] |
11.8794 |
|
4 |
[0.70, 1.70] |
4.6547 |
[2.70, 3.50] |
17.5053 |
|
6 |
[-0.15, 0.50] |
1.0392 |
[2.70, 3.40] |
20.5333 |
|
8 |
[0.00, 0.40] |
1.9524 |
[1.30, 2.30] |
6.5394 |
|
10 |
[-0.10, 0.20] |
0.4619 |
[-0.10, 0.60] |
1.4649 |
In Table X (not included in the article), we have reported Cohen's d score along with 95% confidence intervals to assess the size of the effect when comparing the Control with the free IND and the IND-ves, respectively.
Cohen's d score is a widely used statistical measure of effect size. It quantifies the difference between two group means in terms of standard deviations. If Cohen’s d is equal to 0, then the treatment (here, free IND or IND-ves) and control have no differences in effect. A Cohen’s d score that is greater than zero indicates the degree to which one treatment is more effective than the control. A conventional rule is to consider a Cohen’s d of 0.2 as small, 0.5 as medium, 0.8 as large, and above 1 as very large. In Table X, the numbers marked in red and the numbers marked in green correspond to significant differences of free IND and IND-ves, respectively, when compared to Control. This fact is in agreement with the significant differences marked with ‘**’ in Figure 6.
As the size of sample data is small, the assumptions for constructing parametric confidence intervals are not fulfilled. The confidence intervals constructed here are based on a bootstrap method, by generating 10000 bootstrap samples of median differences. The 95% confidence interval corresponds to the 2.5th and 97.5th percentiles of the bootstrapped distribution. This method provides a robust estimate of the 95% confidence interval for the difference in medians, especially when the volume of data is small.
Normality assumptions for statistical tests should be verified and stated.
We inserted the following observations in the article:
In order to assess the differences between the central values of the Control and IND, and respectively, of Control and IND-ves, we have used the Wilcoxon rank sum test, rather than the t-test for independent variables. The latter one relies on the normality assumption of the compared variables, which is not met here. The size of data is too small for the normality assumption to be valid. The Wilcoxon Rank Sum Test (also called the Mann–Whitney U test) is a non-parametric version of the two-sample t-test for independent samples, meaning that it doesn't rely on this assumption. This test it is used to compare the median values of two groups, and it only requires that the observations from both groups are of continuous type and independent of each other. The underlying null hypothesis of this test is that there is no difference between the compared groups. As the name of the test suggests, instead of using the raw data values to calculate the test statistic, it uses the ranks of the data.
Based on the Wilcoxon Rank Sum Test, we have found significant differences between some of the median values of Control and IND, and respectively, of Control and IND-ves, at different time frames. This fact can also be observed in Figure 6, where ‘**’ means significantly different median values from Control at level
The Pearson correlation coefficient (r = 0.9226) is statistically significant significance level 0.001 (p = 0.000143, 95%CI: [0.70, 0.98]).
While the tail-flick test is an established model, additional pain models (e.g., formalin test or mechanical nociception) would strengthen the analgesic evaluation.
Thank you for your comment regarding the use of the tail-flick test. We acknowledge that while the tail-flick test is an established and reliable model for evaluating analgesic effects, we agree that incorporating additional pain and inflammation models, such as the formalin test or mechanical nociception as well as carrageenan-induced paw edema, could provide a more comprehensive assessment. However, we would like to clarify that the results presented here represent only a portion of a broader, ongoing research project. These findings are part of a more complex set of studies, and additional data from various models will be published in subsequent work to provide a fuller evaluation of the analgesic and antiinflammatory effects.
On line 283 present reference for your research.
Thank you for the observation. We have inserted the reference in the text.
On line 287 the zeta potential must be represented with a positive or negative value.
Thank you for the observation. We have inserted the positive value for Zeta potential in the text.
The manuscript does not specify important parameters such as:
Sonication amplitude and duration (e.g., “ultrasonic homogenizer used under controlled amplitude and temperature” is too vague).
Thank you for the observation. We have inserted in the text the details. Ultrasound treatment was performed at 25% amplitude for 10 minutes at 29°C, generating 20,000 kJ of energy, with pulse cycles of 5 seconds on and 2 seconds off to ensure uniform dispersion. The process was conducted in an ice-cooled water bath to prevent overheating and degradation of the components.
Magnetic stirring speed (rpm) and duration for chitosan coating.
Temperature of ethanol evaporation (if applicable).
Thank you for the suggestion. We have inserted the details in the text. The stirring process was maintained at 800 rpm for 20 minutes at 22°C to ensure proper mixing and coating.
Lipid-to-drug ratio and exact concentration of IND in the vesicles.
How was chitosan coating efficiency confirmed?
Was the coating uniform or patchy?
Report EE% and LE% using HPLC or UV-Vis quantification.
Provide a calculation formula for encapsulation efficiency.
We appreciate the your request for clarification.
The lipid-to-drug ratio in the formulated vesicles was optimized to achieve efficient drug encapsulation and sustained release. Specifically, the ratio was determined based on preliminary formulation trials, ensuring optimal vesicle stability and drug-loading efficiency. The efficiency of IND encapsulation in the vesicles was quantified using spectrophotometric analysis, and the results have been included in our previous article published [reference 32 in the manuscript].
Chitosan coating efficiency was previously confirmed using a combination of physicochemical characterization techniques. Zeta potential analysis demonstrated a shift in surface charge from −19.2 mV for uncoated vesicle, to +24 mV for CHIT-coated vesicles, indicating successful electrostatic interaction between chitosan and the vesicle surface. Additionally, FTIR analysis revealed characteristic peaks corresponding to chitosan functional groups, further confirming its presence on the vesicle surface. We have elaborated on this in the same previous article published [3].
In our published study, EE% was determined using UV-Vis spectroscopy ensuring precise and reliable quantification of IND in the vesicles. The EE% of IND was calculated using the following formula: EE% = [(Ai − Ar) × 100]/Ai, where Ai is the initial amount of IND and Ar represents the amount of IND released from the vesicles. FTIR analysis identified characteristic peaks associated with chitosan functional groups, confirming its presence on the vesicle surface. UV-Vis spectroscopy demonstrated an encapsulation efficiency of 85% for IND in the microvesicles.
We have inserted this information in the manuscript.
Discuss mechanistic reasons for gastric protection (e.g., chitosan mucoadhesive properties).
We appreciate the reviewer’s insightful suggestion.
Chitosan is known to have mucoadhesive properties, which enhance its ability to adhere to the gastric mucosa and form a protective layer [WM Ways, 2018]. This adhesion prolongs the retention time of chitosan at the gastric site, reducing direct exposure of the mucosa to irritants and acidic conditions [Ahmad., 2024, Mura, 2022]. Additionally, chitosan has been reported to enhance mucus secretion, which can further strengthen the mucosal barrier against gastric damage [Subramanian, 2022].
Moreover, chitosan exhibits bioadhesive interactions with mucins due to its cationic nature, forming electrostatic bonds with the negatively charged sialic acid and sulfate residues of mucosal glycoproteins [Ahmad et al., 2024]. This contributes to the stabilization of the gastric mucosa and may aid in tissue regeneration. Furthermore, its ability to chelate bile acids and suppress pepsin activity provides an additional protective mechanism by reducing enzymatic degradation of the mucus layer [Collado-González, 2019]. These combined properties of chitosan, mucoadhesion, enhancement of mucus secretion, electrostatic stabilization, and enzymatic inhibition, are key factors contributing to its gastric protective effects.
We have included this discussion, along with the relevant references, in the revised manuscript as requested.
Several key nanoparticle characterization techniques are missing:
Transmission Electron Microscopy (TEM) or Scanning Electron Microscopy (SEM)
No confirmation of chitosan interaction with lipid vesicles- FTIR spectra would confirm chitosan-polymer bonding and IND encapsulation efficiency.
No assessment of IND crystallinity or polymorphic changes.XRD/DSC data would confirm whether IND remains amorphous or crystalline within the vesicles.
We would like to clarify that this article is a continuation of our preliminary research, which previously focused on the characterization of the carrier systems, evaluation of the encapsulation efficiency of indomethacin, in vitro release studies, and hemocompatibility assessments. The results of SEM analysis and the FTIR spectra were presented in our earlier publication [reference 32 in the manuscript]. Since the characterization of the entrapped systems has been thoroughly discussed in that previous work, we have referenced these findings in the manuscript. These aspects were extensively investigated and reported as part of a larger study on novel carrier systems for NSAIDs and their pharmacodynamic effects in pain and inflammation in laboratory animals. In the present manuscript, we have built upon these findings by focusing on in vivo biocompatibility evaluation, the kinetic release profile of IND in blood, and the influence of these systems on nociceptive reactivity in lab animals, rather than repeating the detailed formulation characterization.

Round 2
Reviewer 1 Report
Comments and Suggestions for Authors
The authors addressed most criticisms properly but the text is still embedded in Table 1-4. Please, correct.
Author Response
Distinguished Reviewer,
We sincerely thank you for the valuable feedback and for acknowledging our efforts in addressing the previous concerns. We apologize for the oversight regarding the formatting of Tables 1–4. We have now corrected the issue and ensured that the text is no longer embedded within the tables. We appreciate your careful review and helpful suggestions.
Reviewer 2 Report
Comments and Suggestions for Authors
The manuscript has been revised according to the revision. It is suitable for publication in its current form.
Author Response
Distinguished Reviewer,
We sincerely thank you for the valuable comments and suggestions provided. We have carefully revised the manuscript according to the feedback, and we believe the current version addresses all the concerns raised. We are pleased to hear that the manuscript is now considered suitable for publication in its current form.
Thank you again for your time and constructive input throughout the review process.